# Understanding X-ray absorption in liquid water using triple excitations in multilevel coupled cluster theory

Sarai Dery Folkestad[1,5], Alexander C. Paul [1,5], Regina Paul (Née Matveeva)[1,5], Sonia Coriani [2], Michael Odelius [3], Marcella Iannuzzi [4] & Henrik Koch [1] ✉

X-ray absorption (XA) spectroscopy is an essential experimental tool to investigate the local structure of liquid water. Interpretation of the experiment poses a significant challenge and requires a quantitative theoretical description. High-quality theoretical XA spectra require reliable molecular dynamics simulations and accurate electronic structure calculations. Here, we present the first successful application of coupled cluster theory to model the XA spectrum of liquid water. We overcome the computational limitations on system size by employing a multilevel coupled cluster framework for large molecular systems. Excellent agreement with the experimental spectrum is achieved by including triple excitations in the wave function and using molecular structures from state-of-the-art path-integral molecular dynamics. We demonstrate that an accurate description of the electronic structure within the first solvation shell is sufficient to successfully model the XA spectrum of liquid water within the multilevel framework. Furthermore, we present a rigorous charge transfer analysis of the XA spectrum, which is reliable due to the accuracy and robustness of the electronic structure methodology. This analysis aligns with previous studies regarding the character of the prominent features of the XA spectrum of liquid water.

The structure of liquid water has long been a subject of controversial debate[1,2], and various experimental and theoretical studies have been conducted to elucidate the matter[3–5]. X-ray absorption (XA) spectroscopy has emerged as a valuable tool in this respect, offering insights into the local structure of the molecular environment of liquid water[2,4,6–10]. However, the interpretation of the experimental XA spectra relies heavily on accurate theoretical modeling. Sophisticated theoretical methods are therefore necessary to improve our understanding of the spectral properties of this essential substance[3,11].

Simulation of the XA spectrum of liquid water is challenging for several reasons. The task can generally be divided into two steps: generating a representative set of molecular structures, and accurately modeling the core-excited states. Both steps are associated with their own theoretical challenges. The first step, the generation of molecular structures, is complicated by the need for accurate ab initio molecular dynamics. For example, intermolecular interactions and the delicate nature of nuclear quantum effects (NQE) must be accounted for. Due to its performance-to-cost ratio, density functional theory (DFT) is generally used for this step. The second step relies on a highly accurate description of the electronic structure of the core-excited molecule and its close surroundings. However, achieving this level of accuracy comes at a significant cost, which typically limits the possibilities of studying bulk properties. Once again, DFT has generally been the method of choice. Consequently, significant differences in the

[1]Department of Chemistry, Norwegian University of Science and Technology, NTNU, 7491 Trondheim, Norway. [2]Department of Chemistry, Technical University of Denmark, DTU, 2800 Kongens Lyngby, Denmark. [3]Department of Physics, Stockholm University, 10691 Stockholm, Sweden. [4]Department of Chemistry, University of Zurich, 8057 Zürich, Switzerland. [5]These authors contributed equally: Sarai Dery Folkestad, Alexander C. Paul, Regina Paul (Née Matveeva). ✉e-mail: henrik.koch@ntnu.no

agreement between experimental and theoretical XA spectra of liquid water have been observed[12–14].

Due to its attractive computational efficiency, transition-potential DFT (TP-DFT)[15], has been extensively used to compute XA spectra of liquid water[13,16,17]. However, choosing an appropriate core-hole potential (half-core-hole (HCH)[15], full-core-hole (FCH)[18], excited-state-core-hole (XCH)[12]) is non-trivial and depends on the investigated system[16,19]. In contrast, time-dependent DFT (TDDFT) has rarely been used to simulate XA spectra of liquid water[17,20]. Generally, TDDFT is known to systematically underestimate core-excitation energies[21–26]. However, Carter-Fenk et al.[27,28] recently demonstrated that TDDFT can provide an adequate description of core-excited states by explicitly accounting for orbital relaxation. Despite the considerable improvement observed using this electron-affinity TDDFT (EA-TDDFT), significant discrepancies with experiment still remain[27].

Another approach that has been used to study the XA spectrum of liquid water[14,29,30] derives from the Green's function framework for the Bethe–Salpeter equation (BSE)[31] using the GW approximation[32–34]. This GW-BSE method[35–37] found its original application in the field of solid state physics[38]. While the GW approximation provides a significant correction to the quasi-particle states, BSE can describe excitonic effects accurately. In the recent study by Tang et al.[30], an excellent match of simulated and experimental spectra are reported. Their approach includes the use of self-consistent quasi-particle wave functions and approximate inclusion of the coupling between the core and high-lying valence excitations. In this way, increased pre- and main-edge intensities are obtained, compared to more standard GW-BSE calculations excluding these effects.

Coupled cluster theory offers a highly accurate description of electronic excitations and spectroscopic properties of molecular systems. Such accuracy comes at a great expense, due to the steep polynomial scaling[39] associated with these models and the molecular system size is therefore severely limited. The most widely used model is the coupled cluster singles and doubles (CCSD) approach. However, it is often necessary to use a more accurate coupled cluster method when modeling X-ray spectroscopy. This is primarily because the excitations from a core orbital result in strong orbital relaxation, which can be captured via, e.g., triple excitations in the wave function[23,40–42].

The high computational cost of the coupled cluster models is the reason for the limited number of such studies on liquid water[3]. In a study by Fransson et al.[17], the X-ray absorption spectra of small water clusters (up to trimers) were computed using Lanczos-based coupled cluster damped linear response theory[40,43]. They used coupled cluster the singles and perturbative doubles (CC2)[44] and CCSD models. List et al.[45] also used a similar response framework together with polarizable embedding[46] to compute the XA spectrum of an extended water system. Here, only a single water molecule was treated quantum mechanically, whereas the rest was described through polarizable embedding. Note that both studies were conducted before the introduction of the core-valence separation approximation in coupled cluster theory[47,48], and hence the size of the water clusters that could be treated was limited. Within the core-valence separation approximation, only core excitations are accounted for when computing the core-level spectra. Neglecting valence excitations while computing the core excitations has been justified by the large energetic and spatial separation between core and valence orbitals[49]. Utilizing the core-valence separation in coupled cluster calculations significantly simplifies simulations of core-excited states.

As delocalized intermolecular excitations are important when describing XA of liquid water, a study where several water molecules are treated with coupled cluster could be of great value. Given its high accuracy, coupled cluster theory can shed light on the composition of the XA spectrum originating from different hydrogen bond configurations present in liquid water. As previously pointed out[50], one challenge in such analyses is the strong reliance on transition dipole moments and transition energies, whose accuracy is limited by the methodology and the system size.

One strategy to reduce the cost of coupled cluster calculations, while retaining high accuracy, is to use a high-level coupled cluster method only in an active region of the molecular system, while describing the rest at a lower level of coupled cluster theory. This approach defines the multilevel coupled cluster (MLCC) methods[51–58]. The entire molecular system is described by a single wave function[53]. The multilevel strategy can be further extended by considering a subset of the inactive orbitals at the mean-field level. This approach is called coupled cluster in Hartree–Fock (CC-in-HF)[57,59]. For the partitioning of the orbital space there are two options to consider. One option is to use localized orbitals in the active region. Alternatively, correlated natural transition orbitals (CNTOs)[55,58,60] can be used. The latter choice generally yields higher accuracy at a lower cost, as the resulting active orbitals are tailored to compactly describe excitations of interest. The degree of (de-)localization of the active orbital space is then determined by the character of the transitions. Active space approaches, such as MLCC and CC-in-HF, are particularly suitable to simulate XA spectra, because core excitations are localized. Theoretical carbon, oxygen and nitrogen K-edge spectra computed using these methods are in excellent agreement with experimental results[53,54,56]. It has also been demonstrated that these methods may be useful for applications to large and solvated systems[57].

In the present work, multilevel coupled cluster methods for core excitations are applied to an extensive sampling of the configuration space to model the XA spectrum of liquid water. State-of-the-art path-integral molecular dynamics (PIMD)[61–63] is used to include nuclear quantum effects. The strongly constrained and appropriately normed (SCAN) DFT functional is employed in the PIMD simulations[64]. We use the multilevel coupled cluster singles, doubles, and perturbative triples in Hartree–Fock (MLCC3-in-HF)[53,58] approach, in addition to the less accurate CCSD in Hartree–Fock (CCSD-in-HF) method to model the core excitations.

In CCSD-in-HF, CCSD is restricted to orbitals localized on the five central water molecules of a water cluster. In the following we refer to this active space as the CCSD space. The remaining orbitals are treated at the Hartree–Fock level; the partitioning of the system is illustrated in Fig. 1. MLCC3-in-HF is built upon CCSD-in-HF, where we choose an additional subset of orbitals which is treated with CC3; see the right panel of Fig. 1. In the following we refer to this additional active space as CC3 space.

## Results and discussion

### The MLCC3-in-HF X-ray absorption spectrum

The XA spectrum of liquid water is usually divided into three regions: a distinctive pre-edge region centered around 535 eV, a main-edge (also referred to as near-edge) region at 537–538 eV, and a post-edge at 540–542 eV[3,10,65]. The pre- and main-edges are the predominant features of the spectrum. The pre-edge peak is attributed to a highly localized excited state and its intensity is ascribed to distorted hydrogen bonds[2,17,66,67] and local liquid disorder[12,14,68]. The main- and post-edges are generally attributed to intermolecular delocalized excitations[20,30].

In Fig. 2, we present the MLCC3-in-HF XA spectrum of liquid water, generated from 45 core excited states for each of the 896 water cluster structures. The spectrum is broadened using Voigt profiles with constant full width at half maximum (FWHM) of ~0.6 eV. Further details on the molecular dynamics trajectory and the generation of the spectra can be found in Methods Section.

We notice a close agreement of the MLCC3-in-HF results with experiment in both absolute energies and spectral shape, especially considering the simplistic broadening scheme employed. The intensity of the pre-edge is slightly underestimated in MLCC3-in-HF. The main-edge feature is relatively narrow and intense, but the post-edge shows excellent agreement with the experimental data. As a consequence,

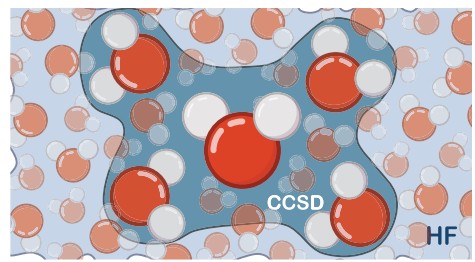
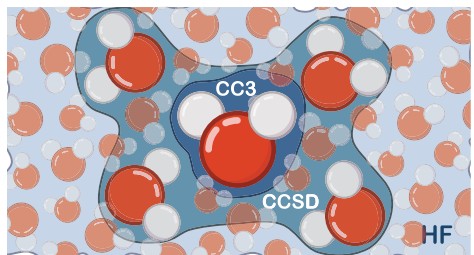

**Fig. 1 | Active spaces in coupled cluster singles and doubles (CCSD-in-HF) and multilevel coupled cluster singles, doubles, and perturbative triples in Hartree–Fock (MLCC3-in-HF).** Illustration of the subsystems used in CCSD-in-HF (left) and MLCC3-in-HF (right). The region containing the five water molecules is referred to as CCSD space, whereas the region around the central water molecule on the right is denoted CC3 space.

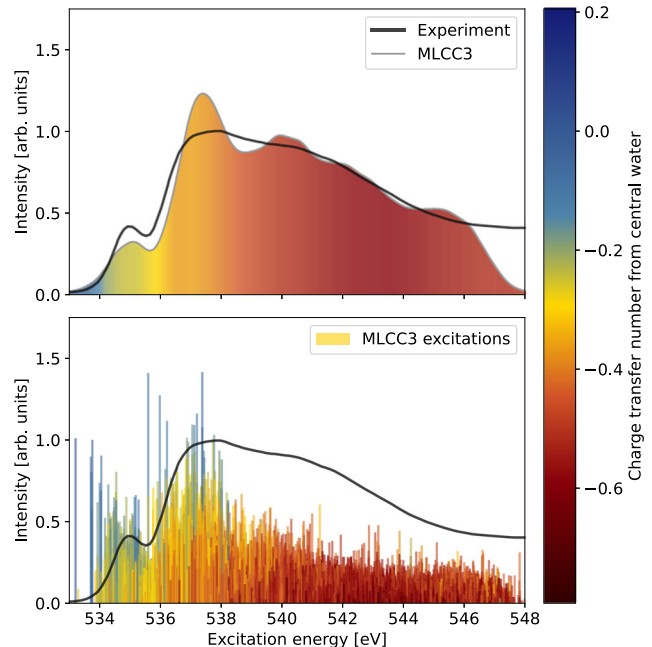

**Fig. 2 | Multilevel coupled cluster singles, doubles, and perturbative triples in Hartree–Fock (MLCC3-in-HF) X-ray absorption spectra with charge transfer analysis.** Upper panel: X-ray absorption spectrum of water clusters at the MLCC3-in-HF level. The color gradient represents the charge transfer character, related to the influence of the environment. Note that the charge transfer character was only calculated for half the snapshots. Lower panel: Stick spectrum for the excitations included in the calculation of the charge transfer character. The oscillator strengths are scaled by 40 to improve visibility. The experimental data was adapted from ref. 65.

the transition from main- to post-edge appears less smooth than the experiment. Note that no shift of the energies is needed to align the MLCC3-in-HF spectrum with the experiment. As shown in Supplementary Fig. 1, this is due to cancellation of errors related to the restriction of triple excitations to the CC3 space. In a full CC3-in-HF calculation, a shift of ~0.5 eV would be needed to match the experimental pre-edge. Possible sources of errors that cancel are the finite basis set, the absence of relativistic effects, and the limited number of orbitals in the CC3 space.

**Charge transfer character of the excitations**

To evaluate the significance of the immediate environment on the XA spectrum, the charge transfer character[69–72] for each transition has been analyzed. The charge transfer number is calculated as the trace of the difference density (i.e., the difference between the excited and ground state densities; for details see Methods section) over the atomic orbitals located on the central water molecule. This partial trace is related to the charge that has been transferred to, or from, the

central water molecule. Hence, it serves as a measure of the interaction with the surroundings. A negative value indicates the removal of electrons while a positive value signifies their addition. The total trace of the difference density is zero by construction.

Half of the water cluster structures were used for the analysis of the charge transfer character. In Fig. 2, we assign a color based on the charge transfer number to each excitation. In the lower panel, we show the individual excitations as bars, while the upper panel displays the averaged charge transfer character for all transitions within a bin of size 0.6 eV. As the excitation energy increases, there is a distinct transition from low to high charge transfer character. This is indicated by the gradual change from positive to increasingly negative values. This means that in the higher energy region the electron gradually moves away from the central water molecule. Because of an artificially reduced density of states (due to the cut-off at 45 excitations), the gradient becomes slightly brighter around 546 eV. The results of our analysis demonstrate that the excitations of the main-edge are a mix of relatively localized transitions (blue) and transitions with some charge transfer character (yellow/orange). Furthermore, they are in line with the expectation that the higher energy excitations will be more diffuse, or delocalized. Accordingly, the results show that the environment becomes particularly important in the post-edge region. The findings of the charge transfer analysis are corroborated by visualizing the difference densities for selected states of a single snapshot, see Supplementary Fig. 3.

Several authors[10,30,45] have suggested that the post-edge is largely characterized by a charge transfer into the second solvation shell and beyond. As mentioned in the Introduction, the coupled cluster calculations presented here are performed in the CCSD space (see Fig. 1). The calculated transitions are thus limited to this active space, and excitations into the second solvation shell are enabled only through diffuse components in the active orbital space. To assess how large these components are, we have plotted the densities from the active occupied and virtual orbitals from the coupled cluster calculations (panel B of Fig. 3). The water molecules of the second solvation shell have some virtual density components, but the density is mainly confined to the first solvation shell. The quality of the MLCC3-in-HF spectra in relation to the experiment (Fig. 2) indicates that such a description is sufficient to accurately capture the post-edge feature. Additionally, we investigated the charge transfer from the first solvation shell into the other shells, visualized in panel A of Fig. 3. Our result underpins findings of previous studies that it is important to go beyond the first solvation shell to simulate the post-edge region. However, we emphasize that the spectrum can be reproduced using a highly accurate electronic structure method, a diffuse basis set on the first solvation shell, and a quantum mechanical embedding for the second solvation shell and beyond.

**Validation of the MLCC3-in-HF calculations**

To further validate the reliability of our results we performed calculations with a larger CCSD space, containing 11 water molecules. Only a

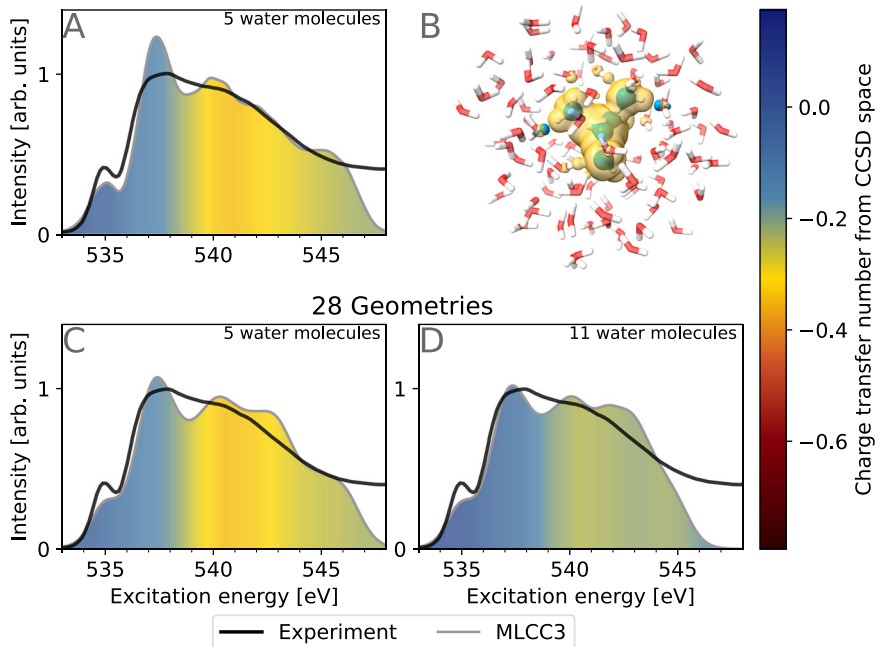

**Fig. 3 | Charge transfer out of the coupled cluster singles and doubles (CCSD) space in the multilevel coupled cluster singles, doubles, and perturbative triples in Hartree−Fock (MLCC3-in-HF) spectrum.** Panel **A**: Charge transfer character out of the CCSD space consisting of 5 water molecules. Panel **B**: Occupied (blue) and virtual (yellow) densities from the active orbitals used in the CCSD space (iso-value 0.1 electrons bohr⁻³). Panel **C**: Charge transfer character out of the CCSD space consisting of 5 water molecules calculated for 28 selected structures. Panel **D**: Charge transfer character out of the CCSD space consisting of 11 water molecules calculated for 28 selected structures. In the panels **C** and **D** we broadened the spectrum using Voigt profiles with 0.2 eV Lorentzian FWHM and 0.5 eV Gaussian standard deviation. The color code here is identical to the one used in Fig. 2. The experimental data was adapted from ref. 65.

small subset of 28 structures was used and the resulting XA spectrum is shown in panel D of Fig. 3 and Supplementary Fig. 2. The comparison of the two active spaces (panel C and D) reveals a small redistribution of intensity from the main- to the post-edge. Additionally, the spectrum is contracted, with the intensity reaching zero at 546 eV due to an increased density of states in the post-edge region. We further observe a strong reduction in the charge transfer number when 11 water molecules are included in the CCSD space. This indicates that most of the interaction with the surrounding water molecules is now contained in the system. Overall, the observed differences in the spectral shape are small and do not justify the drastic increase in computational cost.

To demonstrate the importance of the triple excitations[23] (included in MLCC3-in-HF) for the quality of the obtained XA spectra of liquid water, we performed CCSD-in-HF calculations (shown in Fig. 4). The CCSD-in-HF spectrum was shifted by −1.5 eV to align with experiment. Thus, the absolute peak positions are less accurate than in MLCC3-in-HF. While both models show excellent relative peak positions, the most significant effect of using MLCC3-in-HF is a notable improvement in the relative intensities in the spectrum for the main- and post-edge features. Significant changes of the intensities in CC3 compared to CCSD have been observed previously[73], where CC3 predicts lower oscillator strengths than CCSD.

From Fig. 4, we observe that the pre-edge calculated with CCSD-in-HF is sharper than that of MLCC3-in-HF and its intensity is slightly overestimated compared to experiment. The CCSD-in-HF main-edge is characterized by a narrow peak twice as intense as any other area of the spectrum. Relative to the main-edge, the intensity of the post-edge is underestimated by CCSD-in-HF. In MLCC3-in-HF, this underestimation is corrected due to the smaller oscillator strengths in the main-edge and an increased density of states at high energies compared to CCSD-in-HF. The increased density of states originates from a stabilization of doubly excited states due to the inclusion of triple excitations. Both methods exhibit a peak in the post-edge, above 540 eV, followed by a smooth decrease in intensity up to 545 eV. The excitations

contributing to the post-edge lie above the ionization limit. Therefore a special treatment may be required, e.g., using continuum functions or Stieltjes imaging[17]. However, the quality of the MLCC3-in-HF results indicate that this might not be necessary.

Further analysis of the computed XA spectra reveals that the pre-edge is eliminated when the first excitation is removed from each of the individual spectra (see Supplementary Fig. 4). This excitation is primarily localized on the central water molecule, as evidenced by the low degree of charge transfer shown in Fig. 2. Sampling the contributions to the XA spectrum in the molecular frame of each water molecule, we can decompose the XA intensity into irreducible representations in the $C_{2v}$ point group, see Supplementary Fig. 5. We see that limitations in the description of $B_2$ transitions are predominantly responsible for the too narrow main-edge.

The main spectral regions of the XA spectrum of water have been associated with different hydrogen bond configurations in water[2,18]. However, concerns have been raised regarding the contributions of these configurations. Vaz da Cruz et al.[50] noted that the accuracy of such an analysis relies heavily on the quality of the calculated energies and transition dipole moments, which in turn depends on the chosen electronic structure method. Given that these quantities are highly accurate in MLCC3-in-HF, we have analyzed the spectral composition in terms of various hydrogen bond configurations. First, to determine their distribution we employed the criterion suggested in the Supplementary Information of ref. 2. In the water clusters, 75% of the central water molecules act as double donors, 23% as single donors, and 2% do not participate in any donation. In particular, this corresponds to the following distribution of the most common donor(D)−acceptor(A) hydrogen bond configurations: 53% D2A2, 18% D2A1, 12% D1A2 and 10% D1A1. In Supplementary Fig. 6 we have plotted the averaged spectra for these configurations. Overall, our findings show qualitative similarities with the results obtained by Hetenyi et al.[18] using the FCH DFT approximation. Comparing to the prevalent D2A2 configuration, breaking a donor bond (D1A2) results in a relative

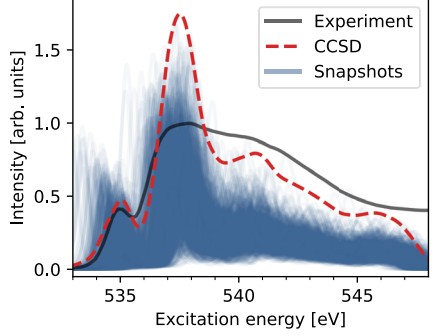
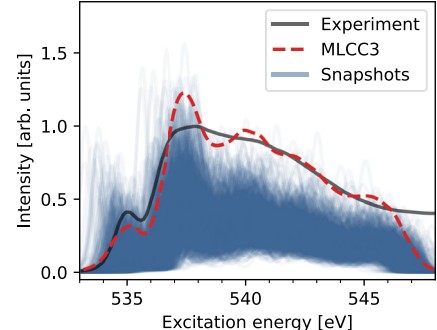
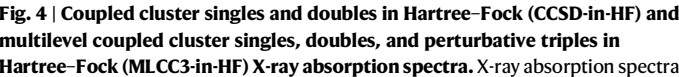

**Fig. 4 | Coupled cluster singles and doubles in Hartree−Fock (CCSD-in-HF) and multilevel coupled cluster singles, doubles, and perturbative triples in Hartree−Fock (MLCC3-in-HF) X-ray absorption spectra.** X-ray absorption spectra of water clusters at the CCSD-in-HF (left) and MLCC3-in-HF (right) levels of theory. The CCSD-in-HF spectrum was shifted by -1.5 eV to match the experiment. The experimental data was adapted from ref. 65.

increase in the intensity of the pre- and main-edges compared to the post-edge region. Breaking an acceptor bond (D2A1), on the other hand, results in a relative increase in the intensity of the pre- and post-edges compared to the main-edge region.

In this work, we have presented the first large-scale application of the multilevel coupled cluster method, MLCC3-in-HF. Because of the high computational cost, standard coupled cluster models have limited applicability for solvated systems, except in combination with embedding approaches. Typically, classical embedding, such as a molecular mechanics or a polarizable continuum embedding, is used. However, the intricate nature of intermolecular interactions of liquid water is responsible for its unique XA spectrum and, therefore, an accurate quantum mechanical description is called for. Our study demonstrates the first successful application of coupled cluster models to the simulation of XA spectroscopy in liquid water. The results have broad implications: enabling accurate simulations of X-ray spectra of solvated chromophores and investigations of the influence of chemical interactions in condensed phase using coupled cluster methods.

Our results indicate that including triple excitations in the coupled cluster wave function, as in MLCC3-in-HF, is paramount to correctly reproduce the experimental intensity ratio between the main- and post-edge. In contrast to CCSD-in-HF, which contains only double excitations, MLCC3-in-HF shows a redistribution of intensities originating from smaller oscillator strengths and an increased density of states in the post-edge region. Due to the inclusion of triple excitations we are also able to reproduce absolute peak positions with unprecedented accuracy.

While the main-edge of the XA spectrum of liquid water has generally been attributed to excitations into the first solvation shell, several authors have suggested that the inclusion of excitations into the second solvation shell or beyond is crucial for the accurate reproduction of the post-edge. We perform the analysis of the charge transfer character for the excitations in the XA spectrum and confirm that the excitations of the main-edge are delocalized over the first solvation shell, whereas the post-edge has contributions beyond it. However, we demonstrate that the effective modeling of the post-edge is possible even without explicitly considering the second or subsequent solvation shells with higher-order electronic structure theory. Our findings reveal that the post-edge can be accurately modeled by a high-level theoretical description, such as MLCC3, of the first solvation shell, while describing the more distant environment using a frozen Hartree−Fock density. One strategy to further improve the accuracy of the computed XA spectrum is to expand the coupled cluster space to match the reported experimental coordination number of 4.7 water molecules[74–76]. Calculations on a subset of the cluster structures, including 11 water molecules in the coupled cluster region, show only a

slight reduction in the intensity of the main-edge, and do not justify the significant increase in computational cost.

## Methods
### Molecular dynamics
The quality of the calculated XA spectra relies significantly on the quality of the sampled structures. Molecular dynamics simulations often assume classical nuclei, but nuclear quantum effects (NQEs) can be important, especially for the description of systems containing light atoms[77]. Liquid water is a notable example where it has been shown that NQEs may influence its structural and dynamical properties[78]. Path-integral ab initio molecular dynamics (PIMD) incorporates NQEs in static properties, which can be observed in the qualitative changes of the radial distribution functions compared to those obtained from standard classical trajectories[62,77]. The effects of the NQEs introduced by PIMD on the XA spectra are well illustrated by the comparison in Supplementary Fig. 7, where linear response TDDFT has been applied to compute spectra on both classical and path-integral trajectories.

The molecular structures used to calculate the XA spectra were extracted from a PIMD trajectory. The PIMD simulations were performed using the i-PI code[79] together with CP2K[61], where the electronic structure is calculated with DFT, using the SCAN functional. These simulations were performed in the canonical ensemble at $T = 300$ K for a system of 32 water molecules under periodic boundary conditions (box size $L_{box} = 9.85$ Å). The volume was held constant such that $\rho = 1$ g cm$^{-3}$. The equations of motion were propagated for 35 ps using a time step $\Delta t = 0.5$ fs. Nuclear quantum effects were included with a thermostated ring polymer contraction scheme using 24 replicas[80].

Structural properties of the resulting model are extensively discussed in a recent study by Herrero et al.[81]. This model is consistent with the tetrahedral coordination of liquid water and provides an accurate description of the structural and dynamical properties within the considered temperature range[81].

### Molecular structure sampling and preparation
For a selection of 28 independent snapshots, water cluster structures were obtained by centering a sphere of 9 Å radius on each of the 32 unique water molecules of the unit cell. This procedure results in a total of 896 water cluster geometries, each containing approximately 100 molecules. The structures provide a 75:23:2 ratio of double-donor, single-donor, and non-donor hydrogen bond configurations of the central water molecule. Calculations at the linear response TDDFT level confirm that our selection of snapshots extracted from the trajectory of a single bead is sufficient to converge the XA spectrum. In Supplementary Fig. 7, we compare the TDDFT[82] XA spectra for different selections of snapshots.

## Electronic structure methods

In coupled cluster theory[39], the wave function is defined as

$$|CC\rangle = e^T |HF\rangle, \qquad (1)$$

where $|HF\rangle$ is the Hartree–Fock reference determinant. The cluster operator $T$ generates excitations of the reference determinant and can be ordered according to the excitation levels of the operator,

$$T = T_1 + T_2 + T_3 + \dots, \qquad (2)$$

where $T_1$ generates single excitations, $T_2$ generates double excitations, and so on. Truncation of $T$ defines the hierarchy of standard coupled cluster methods. For example, in CCSD, $T$ is truncated after $T_2$. High excitation orders can be approximated using perturbation theory. The CC3 model[83] is a notable example where triple excitations are included perturbatively. The multilevel coupled cluster wave function is defined as in Eq. (1). However, excitations of higher order (triple excitations in MLCC3[53] are restricted to an active orbital space. In MLCC3[53], the triple excitations are restricted to an active space.

The CC-in-HF models are quantum mechanical embedding approaches where the density of the target region is correlated with coupled cluster theory, whereas the inactive region is described by a frozen Hartree–Fock density. The orbitals that enter the coupled cluster calculation are chosen by identifying orbitals localized on the target region. In the MLCC3-in-HF and CCSD-in-HF calculations in this work, the active space is defined by the central water molecule and its four nearest neighbor molecules (see CCSD space in Fig. 1). In MLCC3-in-HF, we further partition the orbitals of the target region using CNTOs[60], only a subset is treated at the CC3 level of theory. Additional details on the partitioning can be found in the "Active space selection" section. Note that, although the orbital space treated with CC3 will be largely localized on the central water molecule, it will contain the necessary components to accurately describe excitations into the first solvation shell. It is important to emphasize that in both methods the entire system is characterized by a single wave function.

The coupled cluster models exhibit steep polynomial scaling. The CCSD model scales as $\mathcal{O}(N^6)$, where $N$ relates to the size of the molecular system, and CC3 scales as $\mathcal{O}(N^7)$. The increased scaling, moving from CCSD to CC3, is often limiting, and systems that can be treated routinely with CCSD cannot be treated with CC3. This is the motivation for introducing the MLCC3 approach. CC3 quality can be obtained at a cost approaching that of CCSD, when the active orbital space is appropriate. The power of such an approach is clear from Supplementary Fig. 1. Even with a small active space, the spectral shape obtained in an MLCC3 calculation mirrors that of the full CC3 calculation, while differing significantly from the CCSD spectrum.

For each water cluster, we calculated 45 CCSD-in-HF and MLCC3-in-HF core excited states of the oxygen atom on the central water using core-valence separation[47–49]. This water molecule, and its four closest neighbors, were treated at the coupled cluster level of theory. The remaining water molecules entered the calculation through a frozen Hartree–Fock density. For the central water molecule, we used the aug-cc-pVTZ basis, while aug-cc-pVDZ was used for the other four water molecules in the CCSD space. For the water molecules in the Hartree–Fock space, we adopted the cc-pVDZ basis set[84,85]. The frozen core approximation is applied to all core orbitals except for the oxygen core orbital on the central water. The CC3 space consists of 12 occupied and 72 virtual CNTOs. This active space is sufficient to obtain CC3 accuracy, as can be seen from Supplementary Fig. 1. The calculations were performed using a development version[86] of the $e^T$ program[59].

## Broadening and normalization of theoretical spectra

The theoretical stick spectra are broadened using Voigt profiles with Lorentzian FWHM and Gaussian standard deviation of 0.2 eV. This results in a FWHM of ~0.59 eV. The average spectrum is normalized such that the area underneath the curve matches that of the experiment in the range of 533–545 eV. The experimental data shown in the figures was digitized from Ref. 65 using Web Plot Digitizer[87]. For comparison, Supplementary Figs. 8 and 9 show the spectra broadened using Lorentzian functions with FWHM of 0.2 eV and Gaussian functions with FWHM of 0.6 eV. Despite changing the broadening scheme, the spectral shape, i.e. the main features of the spectrum, remain unchanged.

## Charge transfer analysis

The charge transfer character[69,70] of an excitation in the XA spectrum can be analyzed by considering the density difference between the excited state ($m$) and the ground state (0):

$$\Delta\rho_m = \rho_m - \rho_0. \qquad (3)$$

Since the trace of the one-electron density for an $N$-electron system corresponds to the number of electrons, the trace of $\Delta\rho_m$ is zero.

Using Löwdin population analysis, we can transform $\Delta\rho_m$ from the delocalized molecular orbital (MO) basis to the local orthonormal atomic orbital (OAO) basis,

$$\Delta\rho_m^{\text{OAO}} = \boldsymbol{S}^{\frac{1}{2}} \boldsymbol{C} \, \Delta\rho_m^{\text{MO}} \, \boldsymbol{C}^T \boldsymbol{S}^{\frac{1}{2}}, \qquad (4)$$

where $\boldsymbol{S}$ is the atomic orbital overlap matrix and $\boldsymbol{C}$ the matrix containing the MO-coefficients. It follows that

$$\text{Tr}\left(\Delta\rho_m^{\text{OAO}}\right) = \text{Tr}\left(\Delta\rho_m\right) = 0. \qquad (5)$$

We can now partition the trace of the entire system into the trace of the three subsystems: contributions from the central water molecule (A), the closest neighboring water molecules (B), and the remaining water molecules (C):

$$\text{Tr}\left(\Delta\rho_m^{\text{OAO}}\right) = \sum_{\alpha\in A}[\Delta\rho_m^{\text{OAO}}]_{\alpha\alpha} + \sum_{\beta\in B}[\Delta\rho_m^{\text{OAO}}]_{\beta\beta} + \sum_{\gamma\in C}[\Delta\rho_m^{\text{OAO}}]_{\gamma\gamma} = 0. \qquad (6)$$

The trace of a single subsystem is then interpreted as the number of electrons detached from or attached to this subsystem, depending on the sign.

## Active space selection

Because of the high cost of CCSD and CC3 prohibiting the treatment of the entire system at the coupled cluster level, we employ an active space approach in our study. Two levels of active spaces are introduced: the CCSD and CC3 spaces (see Fig. 1). The CCSD active space is constructed by localizing canonical Hartree–Fock (HF) orbitals for the target region. Semi-localized Cholesky orbitals are used for the occupied space and projected atomic orbitals (PAOs) for the virtual space. The target region (active atoms) contains the central water molecule and the four closest surrounding water molecules of a given cluster, as shown in Fig. 1.

Cholesky orbitals are constructed by a restricted Cholesky decomposition of the occupied Hartree–Fock density, using the atomic orbitals on the active atoms as pivots. To ensure that the active density is localized, diffuse orbitals (orbitals with an exponent smaller than one) were excluded from the pivots. Including diffuse functions will result in an active occupied orbital space that significantly exceeds the target region. By excluding these orbitals from the pivots, a more compact active space is obtained without a significant loss of accuracy in the coupled cluster calculation. For the virtual space, the contribution of the occupied orbitals is projected out of the atomic orbitals centered on the active atoms. These orbitals are the PAOs[88,89]. The final active virtual space is obtained by removing linear dependencies and

orthonormalizing the PAOs using Löwdin orthonormalization. The CCSD active space contains on average 21.4 occupied and 251.0 virtual orbitals. This orbital space is comparable in size to a system consisting of five water molecules, where one molecule is described using the aug-cc-pVTZ basis, and the remaining four molecules with aug-cc-pVDZ and the frozen core approximation. We use correlated natural transition orbitals (CNTOs)[55,60] to determine the CC3 active space (see right panel of Fig. 1). The CNTOs are an extension of the natural transition orbitals, which are generated from a singular value decomposition of single excitation vectors. CNTOs yield compact active spaces for core excitations with MLCC3[58], because they explicitly target the excited states of interest. For the calculations performed in this study, we computed 45 states and used the 12 most important occupied and the 72 most important virtual CNTOs in the CC3 space. An example of the occupied and virtual orbital densities of the CCSD and CC3 spaces can be seen in Supplementary Fig. 10.

### Reporting summary

Further information on research design is available in the Nature Portfolio Reporting Summary linked to this article.

## Data availability

Supplemental figures and details of the computational cost are given in the Supplementary Information. Source data, the water cluster structures, and the data generated and analyzed in this study are available at https://zenodo.org/records/10657213.

## Code availability

The code to run CCSD-in-HF and MLCC3-in-HF calculations and perform the charge transfer analysis is available at https://zenodo.org/records/10837580.

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

## Acknowledgements

This work has received funding from the European Research Council (ERC) under the European Union's Horizon 2020 Research and Innovation Program (grant agreement No. 101020016, H.K.), S.D.F., A.C.P, and H.K. acknowledge funding from the Research Council of Norway through FRINATEK (project No. 275506, H.K.). M.O. acknowledges funding from the Swedish Research Council (grant agreement No. 2021-04521, M.O.), and from the European Union's Horizon 2020 research and innovation programme under the Marie Skłodowska-Curie (grant agreement No. 860553, M.O.). S.C. acknowledges support from the Independent Research Fund Denmark–Natural Sciences, DFF-RP2 (grant No. 7014-00258B, S.C.). We acknowledge computing resources through UNINETT Sigma2 –the National Infrastructure for High Performance Computing and Data Storage in Norway (project No. NN2962k, H.K.), from DeIC – Danish Infrastructure Cooperation (grant No. DeiC-DTU-N3-2023027, S.C.), and from the Swiss National Supercomputing Centre (project ID uzh1, M.I.).

## Author contributions

S.D.F., A.C.P., R.P., S.C., M.O. and H.K. conceived the study. S.D.F., A.C.P., R.P. performed simulations, developed methods to analyze and visualize the data and wrote the initial draft. M.I. performed simulations and analyzed the data. M.O. and H.K. directed the study. All authors discussed the science and the paper.

## Funding

## Competing interests

The authors declare no competing interests.
