## [Peer Review File · Nature Communications]

Enhanced understanding of X-ray absorption in liquid water using novel coupled cluster methodologiesREVIEWER COMMENTS

Reviewer #1 (Remarks to the Author):

The reviewed manuscript, "Enhanced understanding of X-ray absorption in liquid water using novel coupled cluster methodologies" by Henrik Koch and coworkers, presents a new set of simulations of X-ray absorption spectra of liquid water. The paper offers two main findings: (i) triple excitations are important to capture the absolute position of the spectra as well as relative intensities, (ii) the absorption from the core level is a non-local process involving charge transfer. Using path-integral sampling of liquid water, the authors were able to model the spectra with a high level of accuracy.

I enjoyed reading the manuscript; the conclusions were clear, and the science behind the simulations was sound. The manuscript would benefit from a better focus on our understanding of water stemming from the present simulations. Even the title suggests somewhat incremental progress and lacks concreteness. Apart from this general comment, I would ask the authors to focus on specific points as summarized below.

The authors use a Lorentzian broadening of 0.5 eV. This number seems to be chosen arbitrarily, as is often done. However, in the present case, it is not sufficient to select a broadening parameter that merely looks good. Different approaches are compared, and the choice of the broadening parameter affects the shape of the spectra. The Lorentzian shape implies that the value of 0.5 eV should cover the lifetime broadening. In this case, the value is too high. The larger value is justified based on statistical reasoning, yet it should depend on the sample size (and the kernel function should not exclusively be the Lorentzian).

In the introduction, the authors discuss different approaches for calculating liquid water XAS. In some cases, they note very good agreement of the overall shape of the spectrum with the need to shift the peak position. Shifting the absolute position may be unsatisfactory for a theorist, but it can be physically justified if the final spectrum parameter free, e.g., by calculating a well-specified excited state for the monomer.

The authors model liquid water with the SCAN functional. The use of this approach dictates the use of a very small water box of 32 water molecules. While I do not consider myself an expert on water models, I would be concerned about computational artifacts in the liquid structure with such a small box. An excellent alternative to replace costly and sometimes unreliable DFT calculations for ground state sampling is the q-TIP4P model, as seen in J. Chem. Phys. 131, 024501 (2009).

The authors should explicitly discuss the new features emerging from the CC3 calculations, particularly in comparison to the GW-BSE ones, which are probably now considered state-of-the-art calculations. Personally, the CC approach presented here is more appealing to me, but an explicit comparison would fit well in the paper.

As a minor note, I found the discussion of the DFT approaches on page 3 confusing, as it mixes DFT issues in the ground state with the problems of the excited state calculations.

The authors include nuclear quantum effects within the PIMD simulations. These approaches started to be used in condensed phase electronic spectroscopies in the last decade and have become an important part of the story. Perhaps this aspect should be discussed as an additional point in the main text. Some thoughts on the subject are now presented in Figure A7.

Some time ago, an extensive discussion on the structure of liquid water based on XAS interpretation emerged. Here, the authors show that good agreement with the experiment is achieved with water that is probably very much consistent with the standard tetrahedral model. Discussing this part should be more explicit, but to do that, the structure of the liquid water considered in the present work should be more extensively described (probably in the SI, i.e., at least the RDFs should be shown).

Reviewer #2 (Remarks to the Author):

Manuscript by Folkestad et al. presents a theoretical study of X-ray absorption spectrum (XAS) of liquid water using high-level coupled cluster methodologies. The simulations show a good agreement with experimental results when using the highest level of theory (MLCC3-in-HF). Overall, this is a high-quality theoretical work reporting several important results. First, neglecting the effect of triple excitations in coupled cluster calculations make the agreement of simulations with experiment significantly worse, indicating that these excitations are important for accurate predictions. Second, using the methodology that is free of self-interaction errors, the authors quantify the extent of charge transfer character in the XAS of liquid water, showing that it is increasing from pre-edge to post-edge. Third, the authors evaluate the effect of second solvation shell on the post-edge XAS, suggesting that excluding it in coupled cluster calculations does not significantly affect their accuracy. I believe that this paper will be of interest to the readers of Nature Communications, but the authors should address the following points before the paper can be published:

- 1) The paper in its current form is written in a highly technical language that can be difficult to follow even for specialists in this area. Many technical terms used will not be familiar to the readers who do not specialize in quantum chemistry/electronic structure theory. For example, the term "core-valence separation" is used on page 4 without explanation of what it is. Many other examples can be found throughout the paper. I would recommend the authors to significantly revise the text of the manuscript to make it more accessible to the broader audience of Nature Communications.
- 2) The authors' arguments about the effect of including the second solvation shell on the accuracy of post-edge simulations appear to be rather weak. While it appears that neglecting the second-shell effects in coupled cluster calculations still results in a good agreement with experiment for post-edge, this finding may be due to fortuitous error cancellation. As the authors indicate in the paper, the simulation of post-edge is difficult due to the need for describing continuum electronic states. These states are not accounted for in the reported calculations. It is also possible that the MD simulations are inaccurate when it comes to determining the second-shell structure, which may be affecting the results of coupled cluster calculations. One way to disentangle these effects is to perform multilevel CC calculations with more water molecules included in the active space (beyond the first solvation shell). While it is understandable that these calculations will be significantly more expensive, it appears that it should be possible to run them for at least some snapshots of MD trajectories so that the effect of second solvation shell on computed post-edge excitation energies can be estimated. If the authors find that including the second shell does not indeed affect the post-edge excitation energies, this would significantly strengthen their arguments.
- 3) To generate the structures and trajectories of liquid water the authors use path-integral molecular dynamics with SCAN DFT functional, stating that "the resulting model provides an accurate description of the structural and dynamical properties of liquid water within the considered temperature range". This statement is, however, based on comparing the DFT-computed structural and dynamical properties with experiment. However, the SCAN forces and properties can be significantly different from those calculated using coupled cluster theory. While it is prohibitively expensive to perform ab initio molecular dynamics using coupled cluster theory, for a given trajectory one could compute the forces using multilevel CC and compare them with those from SCAN, which could provide a confirmation that the authors' approach to simulating the dynamics of water is a reasonable one. Without this (or an alternative) test, theoretical simulations combining DFT-based ab initio MD with coupled cluster theory can potentially lead to incorrect interpretations that rely on the agreement between theory and experiment that can benefit from error cancellation.

Reviewer #3 (Remarks to the Author):

In the manuscript by Folkestad and co-workers, the X-ray adsorption (XA) spectrum of liquid water is modelled using an active-space multi-level coupled cluster (MLCC) method. The simulated spectrum agrees reasonably well with experiments in all three edges thanks to an approximate inclusion of triple excitations by MLCC3, which shows clear improvement over the commonly used CCSD model. In addition to the spectrum, the authors also conducted charge transfer analysis to probe the electronic nature underlying the excitations in different edges of the spectrum.

The paper is well-written and the conclusion is reasonably supported by the data presented. I am happy to recommend its publication on Nature Communication, given that the following technical comments will be appropriately addressed.

1. Two local active spaces are used by the authors: one made of LOs + PAOs for the CCSD-in-HF embedding and the other made of CNTOs for CC3-in-CCSD-in-HF embedding. Could the authors report more quantitatively on the error introduced by the two active space choices, respectively? For example, the LOs+PAOs active space was restricted on the nearest four water molecules around a center one, how about the second shell? The authors argued that such contributions are small based on the density of the active space orbitals (Left panel of Fig. 4). However, this argument is a bit cyclic in my opinion, as the authors *constructed* the active space not to include basis functions from atoms in the second shell. In any case, showing some convergence plot (e.g., excitation energies, oscillator strengths as a function of number of orbitals/atoms involved in the active space) for selected configuration will make it much more convincing. Similar applies to the active space for CC3.

2. Related to 1, could the authors report the typical computational cost of such calculations? I assume that parallelism is not a problem here since different configurations are completely independent. So reporting the timing/memory cost for each single configuration would be very helpful.

3. The authors mentioned previous works using a damped linear response CC approach, which in my understanding is a direct computation of the spectral function, i.e., something like $A(\omega) \sim \text{Im} \langle 0 | a^\dagger a (H - \omega + i\eta)^{-1} a^\dagger a | 0 \rangle$. Could the authors comment on the advantage of the sum-over-state approach used in this work compared to this direct approach? (assuming CVS can be applied to the spectral approach as well; or is that the main bottleneck?)

Minor points:

4. Some figures and text do not match. E.g., on page 7, "The MLCC3-in-HF spectrum is shown in the lower panel of Figure 1." where there's no "lower panel" of fig 1.

Reply to reviewers

Manuscript ID: NCOMMS-23-36011

Enhanced understanding of X-ray absorption in liquid water using novel coupled cluster methodologies

December 14, 2023

We thank the reviewers for their careful reading of our manuscript and their constructive comments and criticisms. Below, we reply point-wise to their comments and summarize the changes made. All page and equation numbers in our comments refer to the revised manuscript.

All changes made have been highlighted in red in the marked version of the manuscript attached as “Supporting Information for Review Only” document.

Response to Reviewer #1

1. **Comment:** *The reviewed manuscript, “Enhanced understanding of X-ray absorption in liquid water using novel coupled cluster methodologies” by Henrik Koch and coworkers, presents a new set of simulations of X-ray absorption spectra of liquid water. The paper offers two main findings:*

- (i) *triple excitations are important to capture the absolute position of the spectra as well as relative intensities*
- (ii) *the absorption from the core level is a non-local process involving charge transfer.*

Using path-integral sampling of liquid water, the authors were able to model the spectra with a high level of accuracy.

*I enjoyed reading the manuscript; the conclusions were clear, and the science behind the simulations was sound. **The manuscript would benefit from a better focus on our understanding of water stemming from the present simulations. Even the title suggests somewhat incremental progress and lacks concreteness.** Apart from this general comment, I would ask the authors to focus on specific points as summarized below.*

Reply: We thank the reviewer for the comment. We have restructured the manuscript to emphasize the novel features of our study:

- (a) We present the first successful simulation of the XA spectrum of liquid water using highly accurate coupled cluster models, in particular, showing the necessity of triple excitations in the cluster operator.

- (b) We present a new way to analyze the charge transfer character of the different regions of the spectrum. This analysis shows that the first solvation shell is sufficient to accurately describe the XA spectrum, given that a high level of theory and an appropriate basis set are used.

Although our results are in line with previously published work, these new theoretical developments provide an unprecedented confidence in the conclusions, unavailable with cheaper (and in particular DFT based) models.

Changes: See the manuscript where the changed paragraphs are highlighted. We have also changed the title to: “*Understanding the X-ray absorption in liquid water: triple excitations in multilevel coupled cluster theory*”

2. **Comment:** *The authors use a Lorentzian broadening of 0.5 eV. This number seems to be chosen arbitrarily, as is often done. However, in the present case, it is not sufficient to select a broadening parameter that merely looks good. Different approaches are compared, and the choice of the broadening parameter affects the shape of the spectra. The Lorentzian shape implies that the value of 0.5 eV should cover the lifetime broadening. In this case, the value is too high. The larger value is justified based on statistical reasoning, yet it should depend on the sample size (and the kernel function should not exclusively be the Lorentzian).*

Reply: We thank the reviewer for their constructive feedback which we have incorporated in the figures. While we agree that using Voigt profiles is more appropriate, we disagree that the shape of the spectrum depends on the broadening scheme used. Comparing Figure 4 (Voigt profiles with fwhm ~ 0.59 eV), Figure A8 (Lorentzian broadening 0.2 eV fwhm), and Figure A9 (Gaussian broadening 0.6 eV fwhm), we argue that the overall shape of the spectrum is the same. Some details are, however, more or less smooth depending on the broadening scheme. As the reviewer points out, choosing a broadening scheme is difficult and can change the shape of the spectrum substantially. Therefore, we think it is a benefit of our study that the spectra do not significantly change with the broadening scheme.

Changes: All plots where we previously used Lorentzian broadening have been changed to broadening using Voigt profiles with 0.2 eV fwhm of the Lorentzians and 0.2 eV standard deviation of the Gaussian functions (total width of Voigt profiles ~ 0.59 eV). The lifetime broadening was chosen based on this paper [1] which reports widths of 0.1 eV to 0.5 eV.

3. **Comment:** *In the introduction, the authors discuss different approaches for calculating liquid water XAS. In some cases, they note very good agreement of the overall shape of the spectrum with the need to shift the peak position. Shifting the absolute position may be unsatisfactory for a theorist, but it can be physically justified as the final spectrum parameter free, e.g., by calculating a well-specified excited state for the monomer.*

Reply: We agree with the reviewer that the relative peak positions are the most critical feature of the computed spectrum, together with accurate estimates of the transition strengths.

Changes: We removed the statement about the shift in the introduction.

4. **Comment:** *The authors model liquid water with the SCAN functional. The use of this approach dictates the use of a very small water box of 32 water molecules. While I do not consider myself an expert on water models, I would be concerned about computational artifacts in the liquid structure with such a small box. An excellent alternative to replace costly and sometimes unreliable DFT calculations for ground state sampling is the q-TIP4P model, as seen in J. Chem. Phys. 131, 024501 (2009).*

Reply: The model of liquid water used in this work was extensively characterized and compared to other density functionals commonly used to describe water in a study by Herero *et al.* The current model was tested against force fields simulations using, for example, the TIP4P/2005 water model and has been shown to give very good agreement. Based on these findings, we believe that there are no artifacts associated with a small water box.

We used the SCAN functional PIMD trajectories, as they were already available, and rigorously tested through the previous study. However, we agree that a future comparison to a sophisticated force field approach like the q-TIP4P could be very interesting.

5. **Comment:** *The authors should explicitly discuss the new features emerging from the CC3 calculations, particularly in comparison to the GW-BSE ones, which are probably now considered state-of-the-art calculations. Personally, the CC approach presented here is more appealing to me, but an explicit comparison would fit well in the paper.*

Reply: We thank the reviewer for their comment. As both our CC-in-HF and GW-BSE by Tang *et al.* offer state-of-the-art modelling of the XAS of liquid water, a comparison is warranted. However, these two approaches are very different, making a direct comparison of the theoretical framework cumbersome. Incorporating such a discussion into the current work would shift its focus and be too technical. We anticipate that comparing GW-BSE and CC-in-HF would be an interesting topic on its own. Below we highlight some significant differences:

- (a) GW-BSE approach belongs to the family of Green’s function many-body perturbation theories, whereas CC-in-HF is a single-reference wavefunction-based method.
- (b) Tang *et al.* use periodic boundary conditions and plane waves, in our work we consider water clusters with a standard L_2 -basis.
- (c) CC-in-HF approach offers systematic ways to improve the results. For example, the description of the electronic structure within the target region can be improved using a higher level coupled cluster method, e.g. MLCCSDT or MLCCSDTQ. Furthermore, one can increase the size of the target region or enlarge the orbital space for a high level coupled cluster method. We have now addressed the last two points in the manuscript and Appendix. In the GW-BSE approach of Tang *et al.*, several approximations are introduced, each of which can be lifted to possibly improve the theoretical setup. However, in our opinion, the coupled cluster framework offers a clearer path to systematic improvement (and conversely to approximations).
- (d) In the GW-BSE approach of Tang *et al.*, it is possible to approximately include contributions from high-lying valence excited states. These states are explicitly projected out in our CC-in-HF approach, where we employ the core-valence separation approximation.

Figure 1: XAS spectra of water clusters at the MLCC3-in-HF level of theory (orange) compared with the XAS spectrum from Fig. 1a of Ref. [2] calculated using G_0W_0 -BSE@sc- $G^{\text{static}}W_0$ (blue), and the experimental spectrum (gray). In the left panel the MLCC3-in-HF excitations are broadened using Voigt profiles with 0.2 eV Lorentzian width and 0.2 eV Gaussian standard deviation (~ 0.59 eV fwhm). In the right panel the Gaussian standard deviation was scaled linearly to 0.8 eV (~ 2.0 eV fwhm) in the range 536 eV to 544 eV.

- (e) The scaling of the approaches differs significantly. Both approaches are associated with significant computational costs, which are difficult to compare if timings do not stem from the same hardware. Our CCSD-in-HF (and MLCC3-in-HF) scale as $\mathcal{O}(N^6)$, where N relates to the size of the coupled cluster target region. The GW-BSE calculations are $\mathcal{O}(M^4)$, where M relates to the size of the full box of 32 water molecules.
- (f) In our calculations we considered 1 bead of PIMD trajectory for 28 Snapshots each containing 32 water molecules. This amounts to 896 water clusters, where excitations needed to generate our CC-in-HF spectra were computed only for the central molecule. Tang *et al.* considered all 8 beads in their PIMD trajectory for 2 snapshots each containing 32 water molecules. Thus, 512 different water molecules have been excited to obtain the spectrum. The 2 snapshots considered by Tang *et al.* were selected based on a score function designed to best represent the average of all snapshots in the PIMD trajectory.
- (g) To illustrate the differences between the experimental spectrum, our MLCC3-in-HF spectrum, and GW-BSE spectrum by Tang *et al.*, we added figure 1 showing the 3 spectra. Both theoretical spectra are in excellent agreement with the experimental one, with the GW-BSE spectrum being smoother despite using less “geometries” and a smaller FWHM (0.4 eV versus 0.59 eV). Even though Tang *et al.* align the pre-edge of the XA spectra obtained from different individual structures, the pre-edge starts to merge with the main-edge. The main edge in our spectrum is, however, too sharp and too intense, but their post edge (around 542 eV) is a bit too intense, while the high energy region at 546 eV has too little intensity. The features visible in the post-edge of our spectrum can be smoothed by using a broadening scheme where the FWHM is linearly scaled with the excitation energies, which matches the experiment very well.

Due to the difficulties in comparisons of the two approaches, we have chosen not to do so

in the manuscript. Nevertheless, we have expanded on the work by Tang *et al.* in the *Introduction*.

Changes: *Another approach that has been used to study the XA spectrum of liquid water [2–4] derives from the Green’s function framework for the Bethe–Salpeter equation (BSE) [5] using the GW approximation [6–8]. This GW-BSE method [9–11] found its original application in the field of solid state physics. [12] Whereas the GW approximation provides a significant correction to the quasi-particle states, BSE can describe excitonic effects accurately. In the recent study by Tang *et al.* [2], an excellent match of simulated and experimental spectra was demonstrated. Their approach includes the use of self-consistent quasi-particle wave functions and approximate inclusion of the coupling between the core and high-lying valence excitations. In this way, increased pre- and main-edge intensities are obtained, compared to more standard GW-BSE calculations excluding these effects.*

6. **Comment:** *As a minor note, I found the discussion of the DFT approaches on page 3 confusing, as it mixes DFT issues in the ground state with the problems of the excited state calculations.*

Reply: We thank the reviewer for their careful reading of our manuscript and their constructive feedback. We have changed the discussion to focus on excited state calculations as follows:

Changes: *Due to its attractive computational efficiency transition-potential DFT (TP-DFT) [13], has been extensively used to compute XA spectra of liquid water [14–16]. However, different core-hole potentials, namely half-core-hole (HCH) [13], full-core-hole (FCH) [17] and excited-state-core-hole (XCH) [18] provide different accuracy. While FCH is considered inappropriate to describe XAS of liquid water, opinions vary as to whether HCH or XCH is more suitable.[14, 19] In contrast to TP-DFT, time-dependent DFT (TDDFT) has been rarely used for the simulation of XA spectra of liquid water. [15, 20] Generally, this method is known to systematically underestimate core-excitation energies [1, 21–25]. However, recently, Carter-Fenk *et al.* [26, 27] demonstrated that TDDFT can provide an adequate description of core-excited states by explicitly accounting for orbital relaxation. Despite the considerable improvement observed using this electron-affinity TDDFT (EA-TDDFT), significant discrepancies with experiment still remain [26].*

7. **Comment:** *The authors include nuclear quantum effects within the PIMD simulations. These approaches started to be used in condensed phase electronic spectroscopies in the last decade and have become an important part of the story. Perhaps this aspect should be discussed as an additional point in the main text. Some thoughts on the subject are now presented in Figure A7.*

Reply: We thank the reviewer for their comment and added a discussion to Figure A7 in the Appendix. Additionally, the following description was added to the methods section:

Changes: *The quality of the calculated XA spectra relies significantly on the quality of the underlying geometries. Assumption of classical nuclei in molecular dynamics leads to neglecting nuclear quantum effects (NQEs), particularly important for the description of systems containing light atoms [28]. Liquid water is a notable example where it has*

been shown that NQEs may influence its structural and dynamical properties [29]. Path-integral *ab initio* molecular dynamics (PIMD) incorporates NQEs in static properties, which can be observed in the qualitative changes of the radial distribution functions compared to those obtained from standard classical trajectories [28, 30]. The effects of the NQEs introduced by PIMD on the XA spectra are well illustrated by the comparison proposed in Fig. A7, where linear response time dependent DFT has been applied to compute spectra on both classical and path-integral trajectories.

8. **Comment:** *Some time ago, an extensive discussion on the structure of liquid water based on XAS interpretation emerged. Here, the authors show that good agreement with the experiment is achieved with water that is probably very much consistent with the standard tetrahedral model. Discussing this part should be more explicit, but to do that, the structure of the liquid water considered in the present work should be more extensively described (probably in the SI, i.e., at least the RDFs should be shown).*

Reply: We thank the reviewer for their comment. We did not discuss the structure of the liquid water model extensively in this study, because the it has been described and discussed in detail in Ref. 31. We clarified this aspect in the main text by adding the following:

Changes: *Structural properties of the resulting model are extensively discussed in a recent study by Herrero et al. [31] This model is consistent with the tetrahedral coordination of liquid water and provides an accurate description of the structural and dynamical properties within the considered temperature range, as demonstrated in Ref. 31.*

Response to Reviewer #2

1. **Comment:** *Manuscript by Folkestad et al. presents a theoretical study of X-ray absorption spectrum (XAS) of liquid water using high-level coupled cluster methodologies. The simulations show a good agreement with experimental results when using the highest level of theory (MLCC3-in-HF). Overall, this is a high-quality theoretical work reporting several important results. First, neglecting the effect of triple excitations in coupled cluster calculations make the agreement of simulations with experiment significantly worse, indicating that these excitations are important for accurate predictions. Second, using the methodology that is free of self-interaction errors, the authors quantify the extent of charge transfer character in the XAS of liquid water, showing that it is increasing from pre-edge to post-edge. Third, the authors evaluate the effect of second solvation shell on the post-edge XAS, suggesting that excluding it in coupled cluster calculations does not significantly affect their accuracy. I believe that this paper will be of interest to the readers of Nature Communications, but the authors should address the following points before the paper can be published:*

Reply: We thank the reviewer for the positive evaluation.

2. **Comment:** *The paper in its current form is written in a highly technical language that can be difficult to follow even for specialists in this area. Many technical terms used will not be familiar to the readers who do not specialize in quantum chemistry/electronic structure theory. For example, the term "core-valence separation" is used on page 4 without explanation of what it is. Many other examples can be found throughout the*

paper. I would recommend the authors to significantly revise the text of the manuscript to make it more accessible to the broader audience of *Nature Communications*.

Reply: We thank the reviewer for the comment. We have restructured the *Results and Discussion* section and limited most of the technical details to the *Methods* section.

- Comment:** *The authors' arguments about the effect of including the second solvation shell on the accuracy of post-edge simulations appear to be rather weak. While it appears that neglecting the second-shell effects in coupled cluster calculations still results in a good agreement with experiment for post-edge, this finding may be due to fortuitous error cancellation. As the authors indicate in the paper, the simulation of post-edge is difficult due to the need for describing continuum electronic states. These states are not accounted for in the reported calculations. It is also possible that the MD simulations are inaccurate when it comes to determining the second-shell structure, which may be affecting the results of coupled cluster calculations. One way to disentangle these effects is to perform multilevel CC calculations with more water molecules included in the active space (beyond the first solvation shell). While it is understandable that these calculations will be significantly more expensive, it appears that it should be possible to run them for at least some snapshots of MD trajectories so that the effect of second solvation shell on computed post-edge excitation energies can be estimated. If the authors find that including the second shell does not indeed affect the post-edge excitation energies, this would significantly strengthen their arguments.*

Reply: We agree with the reviewer that additional CC-in-HF calculations including more water molecules in the coupled cluster region will improve the reliability of our conclusions. We have performed calculations including 11 instead of 5 water molecules in the CCSD space for 28 arbitrarily chosen geometries. The results are shown in Fig. A3 for two different CC3 spaces and for two broadening schemes (one with the default FWHM (0.59 eV), and one with a bigger width to further smooth the spectrum). Despite the significantly increased number of water molecules in the coupled cluster region, the spectra only show a small redistribution of intensity from the main to the post-edge. We conclude that the overestimation of the post-edge intensity (at around 543 eV) is a feature of the selected 28 geometries, as it disappears when all 896 geometries are included in the spectrum for 5 water molecules. Additionally, the spectrum based on calculations containing 11 water molecules is contracted, with the intensity reaching zero at 546 eV due to an increased density of states in the post-edge region. The two spectra with 11 water molecules, but different CC3 space (panel C, E) are virtually equivalent, indicating that the chosen CC3 space is still sufficient even when increasing the number of water molecules.

We also added the charge transfer analysis for the charge transfer from the coupled cluster region into the remaining water molecules for these 28 geometries. The results are shown in panel C and D of Fig. 3 where we see the charge transfer number is significantly reduced when we include 11 water molecules in the active space. Additionally, comparing panel A and C of Fig. 3, we see that the reduction of the number of snapshots does not affect the color gradient. This indicates that the charge transfer analysis is converged with the number of geometries.

Changes: Figures 3 and A3 were added to the manuscript together with a discussion of the data presented.

4. **Comment:** *To generate the structures and trajectories of liquid water the authors use path-integral molecular dynamics with SCAN DFT functional, stating that "the resulting model provides an accurate description of the structural and dynamical properties of liquid water within the considered temperature range". This statement is, however, based on comparing the DFT-computed structural and dynamical properties with experiment. However, the SCAN forces and properties can be significantly different from those calculated using coupled cluster theory. While it is prohibitively expensive to perform ab initio molecular dynamics using coupled cluster theory, for a given trajectory one could compute the forces using multilevel CC and compare them with those from SCAN, which could provide a confirmation that the authors' approach to simulating the dynamics of water is a reasonable one. Without this (or an alternative) test, theoretical simulations combining DFT-based ab initio MD with coupled cluster theory can potentially lead to incorrect interpretations that rely on the agreement between theory and experiment that can benefit from error cancellation.*

Reply: We thank the reviewer for their suggestion on how to evaluate the configurations generated from the PI-AIMD simulations using this coupled cluster approach. Our study is focused on accurate modelling of core-level spectra. For this purpose, we used an existing MD model, which has previously demonstrated an excellent agreement with experimental data, independent from the present investigation [31]. In this regard, the purely DFT based framework for both MD and XA simulations would be inherently more coherent. However, DFT is well-known to give reliable ground state geometries and CC calculations offer advantages in the modelling of electronic excitations. Therefore, we contend that we have chosen a well-motivated reliable approach by integrating DFT based PI-AIMD simulations, using a modern functional as a foundation for reliable structural models, along with a new advanced tool for spectrum simulations.

The evaluation of the forces obtained using the SCAN functional against the forces computed using coupled cluster theory, suggested by the reviewer, would be worth a separate project. However, the computational expense is still likely to be prohibitive for the water clusters considered in this work. Obtaining multilevel coupled cluster energy derivatives is by no means trivial, and such calculations are not implemented in periodic boundary conditions.

Response to Reviewer #3

1. **Comment:** *In the manuscript by Folkestad and co-workers, the X-ray adsorption (XA) spectrum of liquid water is modelled using an active-space multi-level coupled cluster (MLCC) method. The simulated spectrum agrees reasonably well with experiments in all three edges thanks to an approximate inclusion of triple excitations by MLCC3, which shows clear improvement over the commonly used CCSD model. In addition to the spectrum, the authors also conducted charge transfer analysis to probe the electronic nature underlying the excitations in different edges of the spectrum.*

The paper is well-written and the conclusion is reasonably supported by the data presented. I am happy to recommend its publication on Nature Communication, given that the following technical comments will be appropriately addressed.

Reply: We thank the reviewer for their positive feedback.

2. **Comment:** *Two local active spaces are used by the authors: one made of LOs + PAOs for the CCSD-in-HF embedding and the other made of CNTOs for CC3-in-CCSD-in-HF embedding. Could the authors report more quantitatively on the error introduced by the two active space choices, respectively? For example, the LOs+PAOs active space was restricted on the nearest four water molecules around a center one, how about the second shell? The authors argued that such contributions are small based on the density of the active space orbitals (Left panel of Fig. 4). However, this argument is a bit cyclic in my opinion, as the authors *constructed* the active space not to include basis functions from atoms in the second shell. In any case, showing some convergence plot (e.g., excitation energies, oscillator strengths as a function of number of orbitals/atoms involved in the active space) for selected configuration will make it much more convincing. Similar applies to the active space for CC3.*

Reply: We agree with the reviewer that additional CC-in-HF calculations with (a) a larger CC3 active space and (b) more water molecules will improve the reliability of our conclusions. We have therefore studied the effects of (a) and (b) for 28 arbitrarily chosen geometries:

- (a) We have performed calculations where we gradually increased the CC3 active space from the one used in the study to full CC3. The results are shown in Fig. A1. We show that increasing the CC3 active space shifts the spectrum, but that the spectral shape is preserved.
- (b) We have performed calculations including 11 instead of 5 water molecules in the CCSD space for 28 arbitrarily chosen geometries. The results are shown in Fig. A3 for two different CC3 spaces and two broadening schemes (one with the default FWHM (0.59 eV), and one with a bigger width to further smooth the spectrum). Despite the significantly increased number of water molecules in the coupled cluster region, the spectra only show a small redistribution of intensity from the main to the post-edge. We conclude that the overestimation of the post-edge intensity (at around 543 eV) is a feature of the selected 28 geometries, as it disappears when all 896 geometries are included in the spectrum for 5 water molecules. Additionally, the spectrum based on calculations containing 11 water molecules is contracted, with the intensity reaching zero at 546 eV due to an increased density of states in the post-edge region. The two spectra with 11 water molecules, but different CC3 spaces (panels C, E) are virtually equivalent, indicating that the chosen CC3 space is still sufficient even when increasing the number of water molecules.

Changes: The convergence of the MLCC3 space is shown in Fig. A1 and discussed at the beginning of Section 2 and in Section 3. The effect of including more water molecules in the CCSD space is shown in Figure 3 together with the effect on the charge transfer analysis. Additionally, we compare the spectra for 5 and 11 water molecules in A3 for two broadening schemes and two MLCC3 active spaces.

3. **Comment:** *Related to 1, could the authors report the typical computational cost of such calculations? I assume that parallelism is not a problem here since different configurations are completely independent. So reporting the timing/memory cost for each single configuration would be very helpful.*

Reply: The calculations were run on nodes with two Intel(R) Xeon(R) Gold 6138

CPU @ 2.00GHz with 20 threads each. The calculations were run using 20 threads for parallelization. For CCSD-in-HF, the mean peak memory usage was 59.29 ± 3.22 GB and the mean wall time 22.70 ± 5.66 h. For MLCC3-in-HF, the mean peak memory usage was 62.00 ± 0.67 GB and the mean wall time 23.22 ± 5.64 h. The MLCC3-in-HF time does not include the CCSD-in-HF calculation required to construct the CNTO space because we restart from the CCSD-in-HF results. The large standard deviation of the timings stems from the varying load of the cluster we used.

Changes: We added a table containing the above information and a short discussion of the numbers to the Supporting information.

4. **Comment:** *The authors mentioned previous works using a damped linear response CC approach, which in my understanding is a direct computation of the spectral function, i.e., something like $A(\omega) \sim \text{Im}\left(\langle 0| a^\dagger a (H - \omega + i\eta) a^\dagger a |0\rangle\right)$. Could the authors comment on the advantage of the sum-over-state approach used in this work compared to this direct approach? (assuming CVS can be applied to the spectral approach as well; or is that the main bottleneck?)*

Reply: In very general terms, damped linear response allows one to obtain directly the absorption cross-section in arbitrary frequency regions, from calculations of the imaginary part of the complex dipole polarizability.

As already discussed in several published works [32–35], when a high density of states is expected, or a specific frequency region is desired, a damped response approach can be advantageous over approaches where excitation energies and transition strengths (i.e., “stick spectra”) are computed starting from the lowest excitation energy (we assume this is what the referee means with “sum-over-states approach”).

Note, however, that different technical implementations of damped response theory exist. In one of them, linear response equations are solved for complex amplitudes, and the polarizability is calculated directly over a grid of frequency values [32]; in the other, the damped polarizability is effectively reconstructed from a truncated set of approximate eigenstates obtained using a Lanczos solver [36, 37]. The latter is the approach used in the earlier works of Fransson *et al.* [15], and of List *et al.* [38], as a means to reach the region of core excitations, as it exploits the property of Lanczos solver to yield eigenvalues over the entire frequency region [36, 37]. Without the core-valence separation, however, the Lanczos-based damped approach used by Fransson *et al.* [15] and by List *et al.* [38] still required a very large number of Lanczos eigenvectors, which prevented its applicability to large water clusters.

The later implementation of the core-valence separation within CC response theory for different types of solvers (including the Davidson eigenvalue solver [39, 40] used in the current study, the Lanczos eigenvalue solver [41], and the damped linear response solver [34]) has been paramount to removing one of the major bottlenecks encountered when targeting core spectra with CC methods.

Changes: We opted not to comment on the advantages or disadvantages of the damped response approach, as already amply discussed elsewhere and deemed not relevant in the present context. A couple of sentences on the importance of the core-valence separation to enable efficient calculations of core spectra have been added.

5. **Comment:** *Some figures and text do not match. E.g., on page 7, “The MLCC3-in-HF*

spectrum is shown in the lower panel of Figure 1.” where there’s no “lower panel” of fig 1.

Reply: We thank the reviewer for their careful reading of the manuscript.

Changes: We have ensured that figures and text match.

References

- [1] Norman, P., Dreuw, A.: Simulating X-ray spectroscopies and calculating core-excited states of molecules. *Chem. Rev.* **118**(15), 7208–7248 (2018)
- [2] Tang, F., Li, Z., Zhang, C., Louie, S.G., Car, R., Qiu, D.Y., Wu, X.: Many-body effects in the X-ray absorption spectra of liquid water. *Proc. Natl. Acad. Sci.* **119**(20), 2201258119 (2022)
- [3] Vinson, J., Kas, J.J., Vila, F.D., Rehr, J.J., Shirley, E.L.: Theoretical optical and x-ray spectra of liquid and solid H₂O. *Phys. Rev. B* **85**(4), 045101 (2012)
- [4] Chen, W., Wu, X., Car, R.: X-ray absorption signatures of the molecular environment in water and ice. *Phys. Rev. Lett.* **105**(1), 017802 (2010)
- [5] Nakanishi, N.: A general survey of the theory of the Bethe-Salpeter equation. *Prog. Theor. Phys. Suppl.* **43**, 1–81 (1969)
- [6] Hedin, L.: New method for calculating the one-particle Green’s function with application to the electron-gas problem. *Phys. Rev.* **139**(3A), 796 (1965)
- [7] Hedin, L.: On correlation effects in electron spectroscopies and the GW approximation. *J. Phys. Condens. Matter* **11**(42), 489 (1999)
- [8] Reining, L.: The GW approximation: content, successes and limitations. *Wiley Interdiscip. Rev. Comput. Mol. Sci.* **8**(3), 1344 (2018)
- [9] Onida, G., Reining, L., Rubio, A.: Electronic excitations: density-functional versus many-body Green’s-function approaches. *Rev. Mod. Phys.* **74**(2), 601 (2002)
- [10] Leng, X., Jin, F., Wei, M., Ma, Y.: GW method and Bethe–Salpeter equation for calculating electronic excitations. *Wiley Interdiscip. Rev. Comput. Mol. Sci.* **6**(5), 532–550 (2016)
- [11] Vinson, J., Rehr, J., Kas, J., Shirley, E.: Bethe-Salpeter equation calculations of core excitation spectra. *Phys. Rev. B* **83**(11), 115106 (2011)
- [12] Blase, X., Duchemin, I., Jacquemin, D.: The bethe–salpeter equation in chemistry: relations with td-dft, applications and challenges. *Chem. Soc. Rev.* **47**(3), 1022–1043 (2018)
- [13] Triguero, L., Pettersson, L., Ågren, H.: Calculations of near-edge x-ray-absorption spectra of gas-phase and chemisorbed molecules by means of density-functional and transition-potential theory. *Phys. Rev. B* **58**(12), 8097 (1998)

- [14] Leetmaa, M., Ljungberg, M.P., Lyubartsev, A., Nilsson, A., Pettersson, L.G.M.: Theoretical approximations to X-ray absorption spectroscopy of liquid water and ice. *J. Electron Spectros. Relat. Phenomena* **177**(2-3), 135–157 (2010)
- [15] Fransson, T., Zhovtobriukh, I., Coriani, S., Wikfeldt, K.T., Norman, P., Pettersson, L.G.M.: Requirements of first-principles calculations of X-ray absorption spectra of liquid water. *Phys. Chem. Chem. Phys.* **18**(1), 566–583 (2016)
- [16] Iannuzzi, M.: X-ray absorption spectra of hexagonal ice and liquid water by all-electron Gaussian and augmented plane wave calculations. *J. Chem. Phys.* **128**(20), 204506 (2008)
- [17] Hetényi, B., De Angelis, F., Giannozzi, P., Car, R.: Calculation of near-edge x-ray-absorption fine structure at finite temperatures: Spectral signatures of hydrogen bond breaking in liquid water. *J. Chem. Phys.* **120**(18), 8632–8637 (2004)
- [18] Prendergast, D., Galli, G.: X-ray absorption spectra of water from first principles calculations. *Phys. Rev. Lett.* **96**(21), 215502 (2006)
- [19] Martelli, F.: *Properties of Water from Numerical and Experimental Perspectives*. CRC Press, Taylor & Francis Group, Boca Raton (2022)
- [20] Brancato, G., Rega, N., Barone, V.: Accurate density functional calculations of near-edge X-ray and optical absorption spectra of liquid water using nonperiodic boundary conditions: the role of self-interaction and long-range effects. *Phys. Rev. Lett.* **100**(10), 107401 (2008)
- [21] Besley, N.A., Asmuruf, F.A.: Time-dependent density functional theory calculations of the spectroscopy of core electrons. *Phys. Chem. Chem. Phys.* **12**(38), 12024–12039 (2010)
- [22] Besley, N.A.: Density functional theory based methods for the calculation of X-ray spectroscopy. *Acc. Chem. Res.* **53**(7), 1306–1315 (2020)
- [23] Rankine, C.D., Penfold, T.J.: Progress in the theory of x-ray spectroscopy: From quantum chemistry to machine learning and ultrafast dynamics. *J. Phys. Chem. A* **125**(20), 4276–4293 (2021)
- [24] Bussy, A., Hutter, J.: First-principles correction scheme for linear-response time-dependent density functional theory calculations of core electronic states. *J. Chem. Phys.* **155**(3), 034108 (2021)
- [25] Besley, N.A.: Equation of motion coupled cluster theory calculations of the X-ray emission spectroscopy of water. *Chem. Phys. Lett.* **542**, 42–46 (2012)
- [26] Carter-Fenk, K., Cunha, L.A., Arias-Martinez, J.E., Head-Gordon, M.: Electron-affinity time-dependent density functional theory: Formalism and applications to core-excited states. *J. Phys. Chem. Lett.* **13**(41), 9664–9672 (2022)
- [27] Carter-Fenk, K., Head-Gordon, M.: On the choice of reference orbitals for linear-response calculations of solution-phase K-edge X-ray absorption spectra. *Phys. Chem. Chem. Phys.* **24**(42), 26170–26179 (2022)

- [28] Markland, T.E., Ceriotti, M.: Nuclear quantum effects enter the mainstream. *Nat. Rev. Chem.* **2**(3), 0109 (2018)
- [29] Chen, W., Ambrosio, F., Miceli, G., Pasquarello, A.: Ab initio electronic structure of liquid water. *Phys. Rev. Lett.* **117**(18), 186401 (2016)
- [30] Ceriotti, M., Fang, W., Kusalik, P.G., McKenzie, R.H., Michaelides, A., Morales, M.A., Markland, T.E.: Nuclear quantum effects in water and aqueous systems: Experiment, theory, and current challenges. *Chem. Rev.* **116**(13), 7529–7550 (2016)
- [31] Herrero, C., Pauletti, M., Tocci, G., Iannuzzi, M., Joly, L.: Connection between water’s dynamical and structural properties: Insights from ab initio simulations. *Proc. Natl. Acad. Sci.* **119**(21), 2121641119 (2022)
- [32] Kauczor, J., Norman, P., Christiansen, O., Coriani, S.: Communication: A reduced-space algorithm for the solution of the complex linear response equations used in coupled cluster damped response theory. *J. Chem. Phys.* **139**(21), 211102 (2013)
- [33] Fedotov, D.A., Coriani, S., Hättig, C.: Damped (linear) response theory within the resolution-of-identity coupled cluster singles and approximate doubles (RI-CC2) method. *J. Chem. Phys.* **154**(12), 124110 (2021)
- [34] Faber, R., Coriani, S.: Core–valence-separated coupled-cluster-singles-and-doubles complex-polarization-propagator approach to x-ray spectroscopies. *Phys. Chem. Chem. Phys.* **22**, 2642–2647 (2020)
- [35] Andersen, J.H., Coriani, S., Hättig, C.: Efficient protocol for computing mcd spectra in a broad frequency range combining resonant and damped cc2 quadratic response theory. *J. Chem. Theory Comput.* **19**(17), 5977–5987 (2023)
- [36] Coriani, S., Christiansen, O., Fransson, T., Norman, P.: Coupled-cluster response theory for near-edge x-ray-absorption fine structure of atoms and molecules. *Phys. Rev. A* **85**, 022507 (2012)
- [37] Coriani, S., Fransson, T., Christiansen, O., Norman, P.: Asymmetric-lanczos-chain-driven implementation of electronic resonance convergent coupled-cluster linear response theory. *J. Chem. Theory Comput.* **8**(5), 1616–1628 (2012)
- [38] List, N.H., Coriani, S., Kongsted, J., Christiansen, O.: Lanczos-driven coupled–cluster damped linear response theory for molecules in polarizable environments. *J. Chem. Phys.* **141**(24), 244107 (2014)
- [39] Coriani, S., Koch, H.: Communication: X-ray absorption spectra and core-ionization potentials within a core-valence separated coupled cluster framework. *J. Chem. Phys.* **143**(18), 181103 (2015) <https://doi.org/10.1063/1.4935712>
- [40] Coriani, S., Koch, H.: Erratum: “Communication: X-ray absorption spectra and core-ionization potentials within a core-valence separated coupled cluster framework” [*J. Chem. Phys.* **143**, 181103 (2015)]. *J. Chem. Phys.* **145**(14), 149901 (2016) <https://doi.org/10.1063/1.4964714> https://pubs.aip.org/aip/jcp/article-pdf/doi/10.1063/1.4964714/15519672/149901_1_online.pdf

- [41] Tenorio, B.N.C., Moitra, T., Nascimento, M.A.C., Rocha, A.B., Coriani, S.: Molecular inner-shell photoabsorption/photoionization cross sections at core-valence-separated coupled cluster level: Theory and examples. *J. Chem. Phys.* **150**(22), 224104 (2019)

REVIEWERS' COMMENTS

Reviewer #1 (Remarks to the Author):

The authors have extensively addressed the issues I raised in my original review, and the manuscript has been improved. I have no major criticism of the manuscript in the present version. My comments are as follows:

Response 2: (comment, no action required) I am unsure whether the authors grasped my point. It wasn't the incorrect functional form of the kernel function, but rather the arbitrary value of the broadening parameter. I argued that while Lorentzian broadening is typically associated with lifetime broadening, the actual value was excessively high, indicating a much shorter core hole lifetime. This has been amended, but the arbitrary choice has been transferred to the Gaussian broadening parameter with $\text{FWHM}=0.48$ eV. The selection can be justified by statistical arguments – the width should then decrease with an increasing number of geometries. The current choice is based on the spectrum's stability against the value used, even though it remains unclear how this is quantified. A more systematic approach is possible, but the procedure is now clarified.

- In response 3, the authors extensively discuss the differences and challenges in comparing GW-BSE with their approach. These comments seem ok and they present a compelling Figure 1. However, the decision not to include the comparison in the manuscript appears unconvincing to me. Listing all the arguments in the text is acceptable, but readers should have an equal opportunity to observe both datasets in a single figure. Both datasets appear good, with the GW-BSE data even looking optically better.

Reviewer #2 (Remarks to the Author):

The authors addressed my comments and I recommend the paper for publication in its current form.

Reviewer #3 (Remarks to the Author):

In the revised manuscript, the authors addressed my technical questions regarding approximations made in the electronic structure methodology. In particular, I was mainly concerned about (i) the size of the local active space the authors used for the MLCC3 calculation and (ii) the contribution from the second-shell of solvation water molecules. These questions are not addressed by the convergence tests shown in Fig. A1 and A3, respectively.

Based on the revised manuscript, I am now happy to recommend its publication on Nature Communications.

Reply to reviewers

Manuscript ID: NCOMMS-23-36011

Understanding X-ray absorption in liquid water using triple excitations in multilevel coupled cluster theory

February 14, 2024

Response to Reviewer #1

1. **Comment:** *The authors have extensively addressed the issues I raised in my original review, and the manuscript has been improved. I have no major criticism of the manuscript in the present version. My comments are as follows:*

Comment: *Response 2: (comment, no action required) I am unsure whether the authors grasped my point. It wasn't the incorrect functional form of the kernel function, but rather the arbitrary value of the broadening parameter. I argued that while Lorentzian broadening is typically associated with lifetime broadening, the actual value was excessively high, indicating a much shorter core hole lifetime. This has been amended, but the arbitrary choice has been transferred to the Gaussian broadening parameter with FWHM=0.48 eV. The selection can be justified by statistical arguments – the width should then decrease with an increasing number of geometries. The current choice is based on the spectrum's stability against the value used, even though it remains unclear how this is quantified. A more systematic approach is possible, but the procedure is now clarified.*

Comment: *In response 3, the authors extensively discuss the differences and challenges in comparing GW-BSE with their approach. These comments seem ok and they present a compelling Figure 1. However, the decision not to include the comparison in the manuscript appears unconvincing to me. Listing all the arguments in the text is acceptable, but readers should have an equal opportunity to observe both datasets in a single figure. Both datasets appear good, with the GW-BSE data even looking optically better.*

Changes: The Figure and the comparison has been included in the Supplementary Information.

Response to Reviewer #2

1. **Comment:** *The authors addressed my comments and I recommend the paper for publication in its current form.*

Response to Reviewer #3

1. **Comment:** *In the revised manuscript, the authors addressed my technical questions regarding approximations made in the electronic structure methodology. In particular, I was mainly concerned about (i) the size of the local active space the authors used for the MLCC3 calculation and (ii) the contribution from the second-shell of solvation water molecules. These questions are now addressed by the convergence tests shown in Fig. A1 and A3, respectively. Based on the revised manuscript, I am now happy to recommend its publication on Nature Communications.*